# Enabling low-drift flexible perovskite photodetectors by electrical modulation for wearable health monitoring and weak light imaging

Yingjie Tang [1,2,6], Peng Jin [3,6], Yan Wang [1,2], Dingwei Li[1,2], Yitong Chen[2,4], Peng Ran[3], Wei Fan[2,4], Kun Liang[1,2], Huihui Ren[2,4], Xuehui Xu[3], Rui Wang [2,5], Yang (Michael) Yang [3] ✉ & Bowen Zhu [2,5] ✉

Metal halide perovskites are promising for next-generation flexible photodetectors owing to their low-temperature solution processability, mechanical flexibility, and excellent photoelectric properties. However, the defects and notorious ion migration in polycrystalline metal halide perovskites often lead to high and unstable dark current, thus deteriorating their detection limit and long-term operations. Here, we propose an electrical field modulation strategy to significantly reduce the dark current of metal halide perovskites-based flexible photodetector more than 1000 times (from ~5 nA to ~5 pA). Meanwhile, ion migration in metal halide perovskites is effectively suppressed, and the metal halide perovskites-based flexible photodetector shows a long-term continuous operational stability (~8000 s) with low signal drift (~$4.2 \times 10^{-4}$ pA per second) and ultralow dark current drift (~$1.3 \times 10^{-5}$ pA per second). Benefitting from the electrical modulation strategy, a high signal-to-noise ratio wearable photoplethysmography sensor and an active-matrix photodetector array for weak light imaging are successfully demonstrated. This work offers a universal strategy to improve the performance of metal halide perovskites for wearable flexible photodetector and image sensor applications.

Flexible photodetectors (FPDs) with excellent photoelectric performance and mechanical flexibility have attracted intensive interest for diverse applications, including wearable health monitoring[1], implantable optoelectronics[2], and artificial vision systems[3–5], etc. Metal halide perovskites (MHPs) are promising candidates for next-generation FPDs owing to their excellent photoelectric properties, low-temperature solution-processed fabrication, facile integration with complementary metal-oxide-semiconductor techniques, and compatibility with flexible substrates[3,6–10]. Recently, MHPs-based FPDs have achieved excellent photoelectric performance, including high-gain, ultrasensitive, wavelength-selective, and ultralong linear dynamic range, comparable to conventional vacuum-processed counterparts[10–12]. However, the high defect density (high dark current) and notorious ion migration (unstable dark current) in polycrystalline MHPs[13–15] hamper their photodetection performance, such as operational stability, signal-to-noise ratio (SNR), and limit of

[1]College of Information Science and Electronic Engineering, Zhejiang University, 310027 Hangzhou, China. [2]Key Laboratory of 3D Micro/Nano Fabrication and Characterization of Zhejiang Province, School of Engineering, Westlake University, 310024 Hangzhou, China. [3]State Key Laboratory of Modern Optical Instrumentation, College of Optical Science and Engineering, Zhejiang University, 310007 Hangzhou, Zhejiang, China. [4]School of Materials Science and Engineering, Zhejiang University, 310027 Hangzhou, China. [5]Institute of Advanced Technology, Westlake Institute for Advanced Study, 310024 Hangzhou, China. [6]These authors contributed equally: Yingjie Tang, Peng Jin. ✉e-mail: yangyang15@zju.edu.cn; zhubowen@westlake.edu.cn

detection (LoD), which are essential for long-term wearable health monitoring and image sensor applications.

Amounts of efforts have been devoted to controlling the defect density and prohibiting ion migration of MHPs, such as composition engineering[16], additive engineering[17,18], interfacial engineering[19,20], and dimension regulation[21–23]. Notably, increasing the resistivity of perovskite has been proven a practical approach to obtaining low dark current[16,24,25]. Nevertheless, it is challenging to endow MHPs with high resistivity comparable to that of commercial low-noise silicon photodetectors (PDs)[26]. Meanwhile, in conventional photoconductive-type PDs, dark and photo-induced currents are conducted in the same path (Fig. 1a). High resistivity will, in turn, deteriorate photoexcited current and sensitivity[9,27,28]. Additionally, owing to the electric potential difference between the signal and ground electrodes (Fig. 1c), ion migration will inevitably occur in the conventional photoconductive-type PDs[13,29].

Herein, we propose a general and effective electrical field modulation strategy to reduce the dark current and suppress the ion migration in the photoconductive-type MHP-based PDs. Compared with the conventional photoconductive-type PDs device structure, a control electrode (CE) is introduced to apply the control voltage (Fig. 1b). The control electrode is directly in contact with the MHP to participate in the carrier transport process, which is different from a field-effect transistor configuration (Supplementary Fig. 1). As there is

no gate dielectric layer, it could provide a more direct and stronger current control capability. Under the electrical field modulation, all the dark current emitted from the ground electrode can be attracted and collected by the control electrode, while the signal electrode can obtain photocurrent without dark current. In addition, the applied control voltage can change the electric field distribution between the signal and ground electrodes (Fig. 1d). Because the potential distribution around the signal electrode is uniform, the ion migration (driven by the electric field) in the area near the signal electrode can be effectively suppressed.

We first theoretically illustrate the process of dark current reduction and ion migration suppression under the electrical field modulation strategy. Additionally, this strategy was proved effective in various scenarios, including different perovskite compositions (MAPbI$_3$, MAPbBr$_3$ and FA$_{0.92}$Cs$_{0.04}$MA$_{0.04}$PbI$_3$), working voltages (0.1, 0.2, and 0.5 V), and thickness of materials (500 nm, 1 μm, 2 μm, and 1 mm). As a result, the dark current of solution-processed MHPs-based FPDs with CE reduced over 1000 times from ~5 nA to ~5 pA under electrical field modulation while maintaining high photocurrent (Fig. 1e, f). In addition, the ion migration near the signal electrode was effectively suppressed, and the FPDs showed excellent operational stability with low signal drift (~4.2 × 10$^{-4}$ pA per second) and ultralow dark current drift (~1.3 × 10$^{-5}$ pA per second) after 8000 s operation. By virtue of their high performance, the MHPs-based FPDs successfully

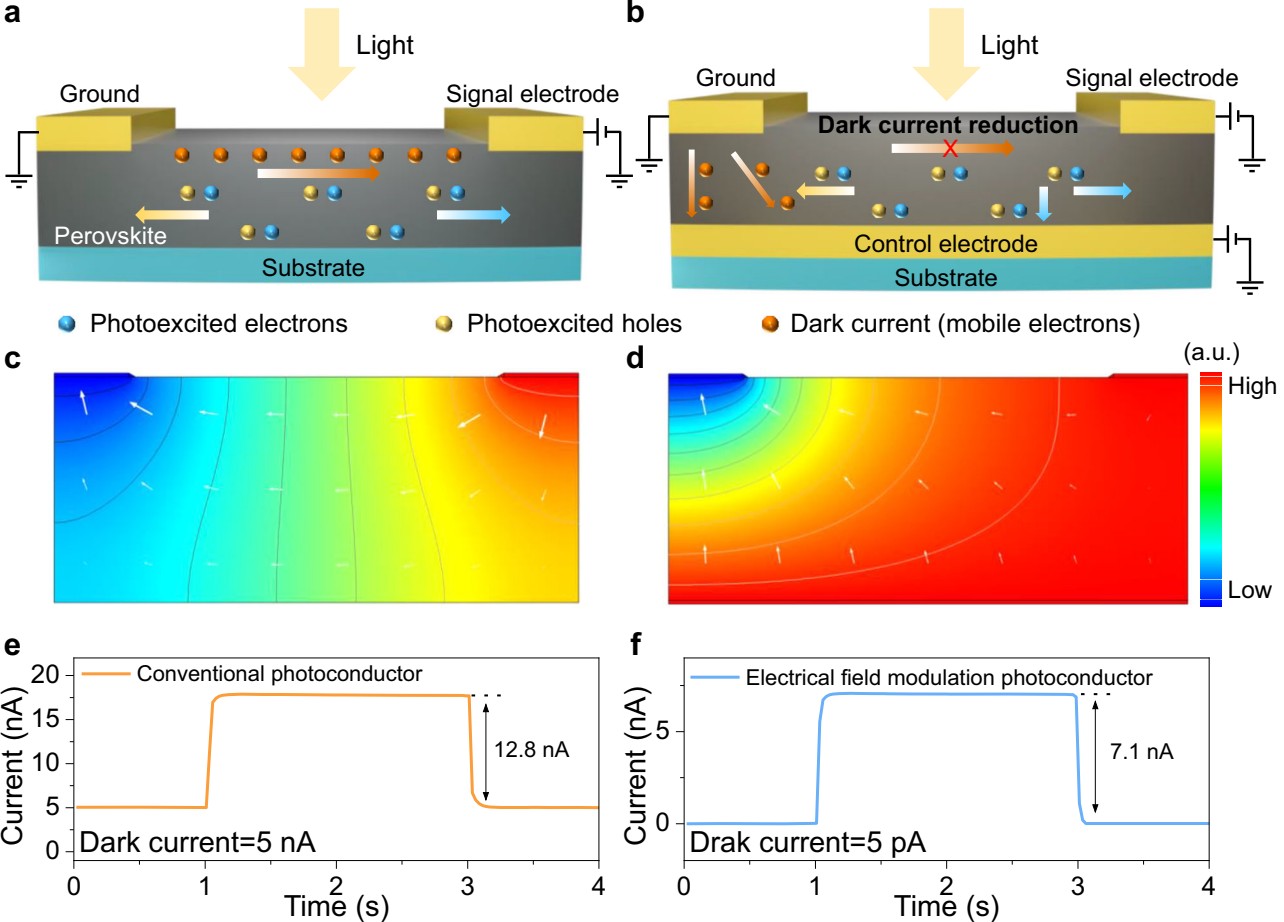

**Fig. 1 | Comparison of conventional photoconductive-type and electrical field modulated flexible photodetectors (FPDs). a** Working mechanism of conventional photoconductive-type FPD. The driving voltage can separate and transport the photoexcited carriers (electrons and holes) in the perovskite film. The photocurrent and dark current are wholly collected by the signal electrode. **b** Working mechanism of electrical field modulated FPD. The dark current emitted from the ground electrode can be attracted by the control electrode. And the signal electrode can collect the photoexcited carriers without dark current. **c**, **d** Electric field distributions in the conventional photoconductive-type and electrical field modulated FPD simulated by the COMSOL software. a.u. arbitrary units. **e**, **f** Time-resolved photocurrent responses of conventional photoconductive-type and electrical field modulated FPD (**f**) under light illumination (520 nm, 100 μW/cm²).

work as a wearable photoplethysmography (PPG) sensor to detect the blood pulse signals with high fidelity under low incident light density (2 mW/cm², 800 nm). Finally, the MHPs-based photodetector with CE was monolithically integrated with a 16 × 16 thin-film transistor (TFT) based flexible active-matrix backplane, capable of imaging weak light distribution with high contrast. This work provides a general approach to achieving high-performance MHPs-based FPDs for wearable and image sensor applications.

## Results

### Working mechanism of the electrical field modulated FPDs

According to the different control voltage values, the working conditions of the electrical field modulated photodetector are divided into four stages when the signal electrode is applied with constant voltage (>0 V) (Fig. 2a). At the dark condition when the control voltage is set as 0 V, the dark current is injected into the perovskite film from the ground electrode and control electrode and transported to the signal electrode due to the potential difference (Fig. 2ai). When the applied control voltage increases (>0 V), the dark current injected from the ground electrode will be partially attracted by the control electrode. Thus, the dark current transported to the signal electrode will be reduced (Fig. 2aii). When the applied control voltage continuously increases, in principle, there is a critical voltage (CV) that enables the dark current—collected by the signal electrode—to be completely reduced to zero (Fig. 2aiii). If the applied control voltage exceeds CV, the dark current will be injected into the perovskite film from the signal electrode and ground electrode and then transported to the control electrode (Fig. 2aiv). As a result, the current signal measured on the signal electrode will be negative. The photoexcited carriers (electrons and holes) transport process in the electrical field modulated photodetector under light illumination is sketched in Supplementary Fig. 2. Although the control electrode will partially attract the photoexcited carriers and weaken the photocurrent collected by the signal electrode when the control voltage is set as CV. The completely reduced dark current will effectively improve the signal-to-noise ratio (SNR) of the signal electrode. To further illustrate the influence of control voltage, we provide the simulation of the potential distribution in perovskite film using COMSOL software (Fig. 2b). The channel length was set as

100 μm and the thickness of the perovskite film was set as 60 μm. When the control voltage is set as CV, the potential distribution near the signal electrode is uniform (Fig. 2biii). Hence, the ion migration near the signal electrode, driven by the potential difference, can be suppressed sufficiently. The corresponding in situ photoluminescence (PL) intensity test around signal electrode also demonstrated the effective suppression of ion migration by electric field modulation strategy (Supplementary Fig. 3). Although there is still ion migration near the ground electrode, it will not affect the signal measured from the signal electrode. In this work, to obtain a high SNR and stable photocurrent signal, the control voltage should be set as CV.

To further verify the effectiveness of the electrical field modulated method in practical application, the MHPs-based ($FA_{0.92}Cs_{0.04}MA_{0.04}PbI_3$, thickness of 500 nm) FPD with CE was fabricated on the colorless polyimide (CPI) substrate. The detailed configuration is shown in Supplementary Fig. 4. When the signal electrode is applied with 0.1 V, the dark current collected by the signal electrode gradually decreases with the increase of control voltage and even reaches negative values (Fig. 2c). As expected, the dark current can be reduced to zero (stage iii) when the control voltage is set as CV. Figure 2d compares the current-time (I-t) curve of the FPDs under the dark condition at a working voltage (signal electrode) of 0.1 V when the CE was applied with 0, 0.05, 0.1 and 0.15 V (stage i to iv). When the working voltage (0.1 V) was applied, and the control voltage was set as 0 V, mobile ions would move in the perovskite film (driven by the potential difference) until reaching the equilibrium state (>2 s), resulting in the high (5 nA) and drifting dark current measured from the signal electrode (stage i)[30]. When the control voltage was set as 0.1 V (critical voltage), the dark current was reduced from -5 nA to -5 pA, and the equilibrium state was reached in only 0.08 s (stage iii), which proves the effectiveness of the electric field modulation strategy for dark current reduction and the ion migration suppression around the signal electrode. In this scenario, the long-term stability of the electric field modulated FPD was also successfully demonstrated (Fig. 2e). After ~8000 s continuous light illumination (520 nm, 50 nW/cm²), the photocurrent and dark current of FPD only drifted from 61.75 to 65.11 pA (-4.2 × 10⁻⁴ pA per second) and from 4.95 to 5.06 pA (-1.3 × 10⁻⁵ pA per second), respectively.

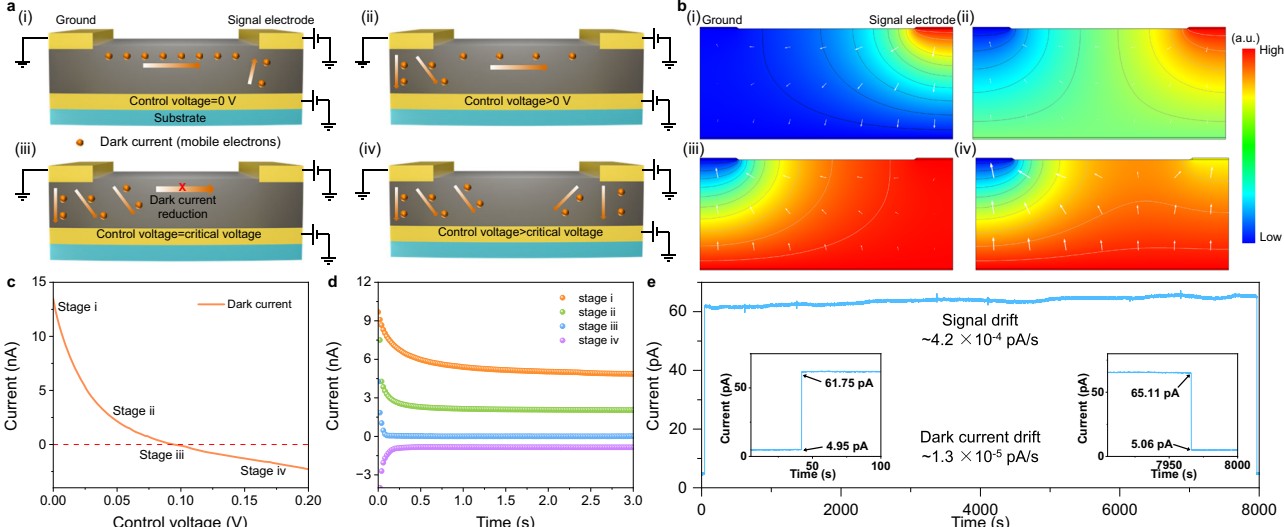

**Fig. 2 | Detailed working mechanism of the electrical field modulated FPD.** **a** Dark current transport process under different control voltage in dark conditions (stage i: control voltage = 0 V, stage ii: control voltage > 0 V, stage iii: control voltage = critical voltage, stage iv: control voltage > critical voltage). **b** Electric field distributions in perovskite film under different control voltages. a.u. arbitrary units. **c** Dark current collected by the signal electrode varies with the control voltage when the signal electrode is set to 0.1 V. **d** I-t curve of the signal electrode under different control voltage. **e** The long-term operational stability of electrical field modulated FPD (at stage iii) under weak light illumination (520 nm, 50 nW/cm²) for ~8000 s. Photocurrent drifted from 61.75 to 65.11 pA after ~8000 s (-4.2 × 10⁻⁴ pA/s). Dark current only drifted from 4.95 to 5.06 pA after ~8000 s (-1.3 × 10⁻⁵ pA/s).

In addition, to determine the generality of the electrical field modulated strategy, we also fabricated MHPs-based FPD with different perovskite compositions (e.g., MAPbI$_3$ and MAPbBr$_3$) and perovskite film thickness (500 nm, 1 μm, 2 μm, and 1 mm). It was demonstrated that the electrical field modulated strategy is applicable to different perovskite materials and different working voltages (0.1, 0.2, 0.5, and 10 V), and the dark current can be reduced to nearly zero when the control voltage was set as CV (Supplementary Figs. 5–10). When the thickness of perovskite is 500 nm, 1 μm, and 2 μm, the value of CV is equal to the working voltage (Supplementary Figs. 7 and 8). When the thickness of perovskite reaches 1 mm, a larger voltage (CV = 35 V) is needed to attract the dark current so that the dark current of the signal electrode (working voltage = 10 V) can be reduced to zero. The simulation of potential distribution in perovskite film of the corresponding thicknesses (500 nm and 1 mm) are shown in Supplementary Figs. 9 and 10. In addition, the roughness and quality of the perovskite film also have an impact on the value of CV because a perovskite surface with a large roughness will affect the distribution of the control electric field and lead to a large local electric field (at sharp region)[31,32], resulting in a reduction in CV value from 100 to 90 mV (Supplementary Fig. 11).

## Photodetection performance of the flexible photodetector

To systematically investigate the photoelectric properties of the electrical field modulated FPD, visible light with different wavelength (405, 520, 600, and 800 nm) were used to illuminate the FPD, as described in Fig. 3a. Compared with the composition of pure FAPbI$_3$ and MAPbI$_3$ perovskite, a more stable cesium-containing triple cation perovskite (FA$_{0.92}$Cs$_{0.04}$MA$_{0.04}$PbI$_3$, thickness of 500 nm) is used as photo-sensing material in this work, which has been proven to be more stable and less affected by fluctuating surrounding variables[33]. The optical properties of the perovskite are investigated by ultraviolet-visible (UV-Vis) spectrophotometer and PL spectrometers (Fig. 3b). The PL spectrum of the perovskite film measured with the excitation of a 532 nm laser exhibits a peak at 803 nm (blue trace), corresponding to a bandgap of 1.54 eV, which is consistent with previous report[34]. And the UV-Vis absorption spectrum of the perovskite (Fig. 1b, orange trace) exhibits an apparent absorption in the visible light region, rendering it feasible for visible light detection. The working voltage was fixed at 0.1 V, unless otherwise specified, to collect the photoexcited signal of the FPD.

To further evaluate the improvement of the FPD photo-sensing performance under CV, the responsivity ($R$) and specific detectivity

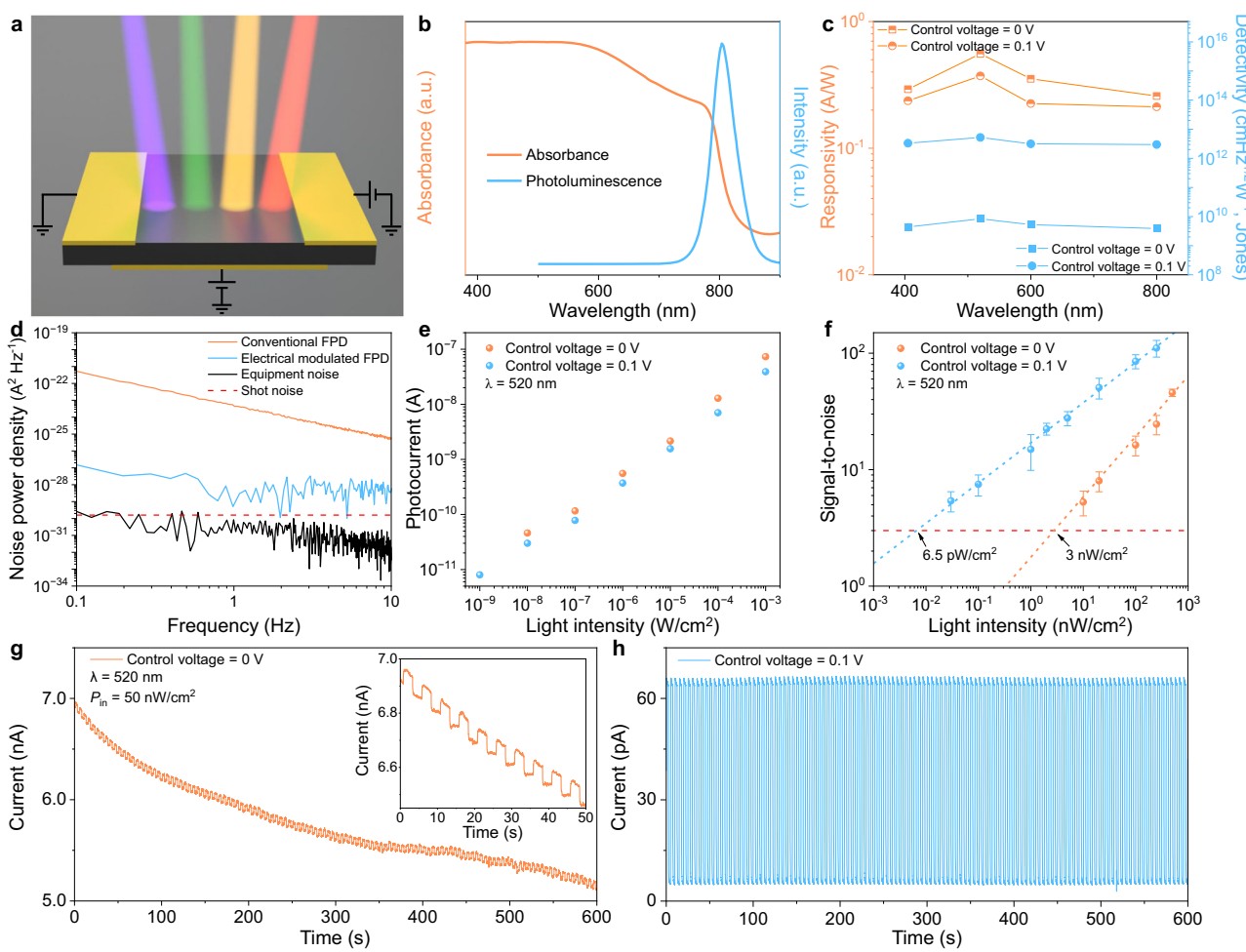

**Fig. 3 | Photoelectric properties of the electrical field modulated FPD when the control electrode was applied with 0 and 0.1 V. a** Sketch image of electrical field modulated FPD under illumination with varied wavelengths from 405 to 800 nm. **b** Absorption spectra (orange trace) and photoluminescence spectra (blue trace) of the perovskite film. a.u. arbitrary units. **c** Dependence of responsivity and detectivity on light wavelength when the light intensity was fixed at 1 μW/cm². **d** Noise power density of the dark current of the FPD when the control electrode was applied with 0 and 0.1 V. The instrument noise floor and calculated shot noise are included for reference. **e** Power-dependent photocurrent under 520 nm light illumination. **f** Signal-to-noise ratio (SNR) of FPDs under different light intensities of 520 nm. The error bars represent standard deviation and are calculated according to variations in the current signal. The red dashed line represents an SNR of 3. The detection limit (SNR = 3) can reach 6.5 pW/cm² when the control electrode was applied with 0.1 V. **g, h** Long-term photocurrent response of the FPDs under 520 nm light (50 nW/cm²) at a working voltage of 0.1 V when the control electrode was applied with 0 V (**g**) and 0.1 V (**h**). Inset in (**g**) shows evident baseline drift indicating the ion migration in the FPDs. The long-term operational stability was realized by applying a 0.1 V control voltage.

($D^*$) of the FPD were extracted using the following equations[9]:

$$R = \frac{I_{photo} - I_{dark}}{P_{in}A} \tag{1}$$

$$D^* = \frac{R\sqrt{AB}}{I_{noise}}$$

Where $I_{photo}$ is photocurrent, $I_{dark}$ is dark current, $P_{in}$ is incident light power density, $B$ is the bandwidth, $I_{noise}$ is the measured total noise, and $A$ is active area. The wavelength-dependent responsivity and detectivity curves with the light fixed at 1 µW/cm² are shown in Fig. 3c. The spectral profile of responsivity peaks at 520 nm, reaching 0.55 (control voltage = 0 V) and 0.37 A W⁻¹ (control voltage = 0.1 V), respectively. Owing to the attractive effect of the control electrode on the photoexcited carriers, the values of $R$ are lower when the control voltage is 0.1 V. Meanwhile, the significant reduction of the dark current lead to an increase in the $D^*$ from $8.3 \times 10^9$ (control voltage = 0 V) to $5.2 \times 10^{12}$ (control voltage = 0.1 V) Jones at 520 nm, indicating a great potential for detecting weak light. And the dependence of $R$ and $D^*$ on the light intensity at different wavelengths (405, 520, 600, and 800 nm) are shown in Supplementary Fig. 12. When the control voltage was set at 0.1 V and incident light intensity was 1 nW/cm², the highest detectivity of FPD reached $1.1 \times 10^{14}$ Jones at 520 nm, outperforming previously reported FPDs[7,35–37]. The values of $R$ and $D^*$ decrease with the growing light intensity, which is attributed to the increased recombination of photoexcited carriers under high light intensity[38]. To avoid performance overestimation, we probed the noise power spectrum of the FPD (Fig. 3d). When the control voltage was set as CV, the measured noise decreased from $4.4 \times 10^{-24}$ to $5.1 \times 10^{-30}$ A² Hz⁻¹ at a frequency of 1 Hz (working voltage = 0.1 V). Figure 3e shows the photocurrent of the FPD under light illumination of various intensities (wavelength of 520 nm), indicating the linear relationship between the incident light density and photocurrent.

LoD is another important performance metric of PDs, which plays a vital role in various applications such as biomedical imaging, environmental monitoring, and communications[39–41]. To estimate the LoD of the PFD, we defined LoD as the light intensity which can produce a signal greater than three times the noise level, meaning the signal-to-noise ratio (SNR) is 3[42]. And the SNR was extracted as:

$$SNR = \frac{\left(\bar{I}_{photo} - \bar{I}_{dark}\right)}{\sqrt{\frac{1}{N}\sum_i^n \left(I_i - \bar{I}_{photo}\right)^2}} \tag{3}$$

Where $I_i$ is the measured photocurrent, $\bar{I}_{photo}$ is the average measured photocurrent, and $\bar{I}_{dark}$ is the average measured dark current. We derived the SNR of FPD under different light intensities (wavelength = 520 nm), as shown in Fig. 3f. The extracted LoD of the FPD is 6.5 pW/cm² at 0.1 V control voltage, which is 460 times lower than that of 3 nW/cm² at 0 V control voltage. Benefiting from the low dark current and high signal current characteristics, the FPD showed prominent optical switching properties (SNR = 6.1) under 520 nm weak light illumination (30 pW/cm²), as shown in Supplementary Fig. 13. Similarly, the dependence of SNR on the light intensity at different wavelengths (405, 600 and 800 nm) is shown in Supplementary Fig. 14. When the control voltage is 0.1 V, the LoD decreases from 29.5, 5, and 15.4 nW/cm² to 52, 37, and 66 pW/cm² at 405, 600, and 800 nm wavelengths, respectively, which proves that the applied control voltage could effectively improve the weak light detection performance of MHPs-based FPDs (Supplementary Fig. 15).

Furthermore, we investigated the operational stability of the FPD under the weak light of 520 nm wavelength (0.2 Hz, 50 nW/cm²). In the

test duration of 600 s with 120 light ON/OFF cycles, the dark current drifted from 6.9 to 5.1 nA (-3 pA per second) when the control voltage is 0 V, as shown in Fig. 3g. Especially during the first 50 s of the test, the dark current drifted at a rate of -10 pA per second (from 6.9 to 6.4 nA), which leads to the low SNR of the signal, as shown in the inset of Fig. 3g. In contrast, the dark current under 0.1 V control voltage modulation, reveals almost no drift, indicating ion migration was effectively suppressed (Supplementary Fig. 16). The photocurrent noise is mainly caused by the input of electric pulse signals from the function generator, which was not observed when a mechanical chopper was used to provide optical pulse signals (Supplementary Fig. 17). The response speed of FPDs was measured by chopper modulated 520 nm light illumination (1 mW/cm²) under different working voltages and control voltages (Supplementary Figs. 18 and 19). With the increase in working voltage, the rise time of FPDs gradually decreases because the increased working voltage reduces the carrier transit time. In addition, the photocurrent also increased with the increase of working voltage, as shown in Supplementary Fig. 20. Meanwhile, the influence of electrodes with different work functions on FPDs is shown in Supplementary Figs. 21–23. Compared with other electrodes, such as Ag or Al, the Au electrode provides superior performance because of good band alignment, low contact resistance and good carrier transport, and chemical stability (Supplementary Fig. 24).

## Application of the FPD for wearable bio-signal detection

To demonstrate the practical application for wearable electronics, we carried out a PPG test based on the FPD. The basic working mechanism of a transmission mode PPG sensor is shown in Fig. 4a. When a light beam with a wavelength of 800 nm is projected onto the skin surface of the fingertip, the light beam will be transmitted to the FPD by transmission, while a part of the light is absorbed, reflected, and scattered by human tissues. During the test process, the light intensity detected by FPD will fluctuate with the changes in the volume of blood vessels caused by heartbeat. Specifically, when the heart is contracting, the increased blood volume in vessels will reduce the light intensity (be absorbed by blood) that can be detected by the FPD. And during diastole, on the contrary, more light can be transmitted to the FPD. Hence, the heart rate (HR) can be extracted from the FPD signals to evaluate the cardiopulmonary function of humans[43,44].

As shown in Fig. 4b, a light beam (800 nm) emitted from the optical fiber was projected onto the fingertip. And the FPD was stuck on the finger pulp by a transparent adhesive film to detect the light intensity changes. To minimize noise signals from the finger movement and surrounding environments, a flexible cable was used to connect the FPD with the external signal acquisition circuit, and the whole measurement was conducted in the dark condition. To investigate the performance improvement of FPD by our proposed method, the PPG tests were conducted with a working voltage of 0.1 V under different incident light intensities (72, 4.6, and 2 mW/cm²) when the control voltage was set to 0 V and 0.1 V, respectively (Fig. 4c–e).

Figure 4c shows the FPD performance under high incident light intensity (72 mW/cm²), and the device in both conditions (with/without control voltage) could successfully obtain high-fidelity PPG signals. When the incident light intensity decreased to 4.6 mW/cm², significant baseline drift appeared in the signal of FPD (control voltage = 0 V), although the periodic blood pulse signal can be distinguished (Fig. 4di). As the incident light intensity was further reduced to 2 mW/cm², the blood pulse signal was overshadowed by the baseline drift and cannot be discriminated (Fig. 4ei). In contrast, high-fidelity PPG signals were acquired under the 0.1 V control voltage modulation regardless of the light intensity (Fig. 4dii and eii). From the measurement results, the blood pulse frequency was calculated to be 67 beats per minute. The above experimental results successfully demonstrate the capability of the FPD to obtain high SNR PPG signals under low light density through electric field modulation, which shows the great

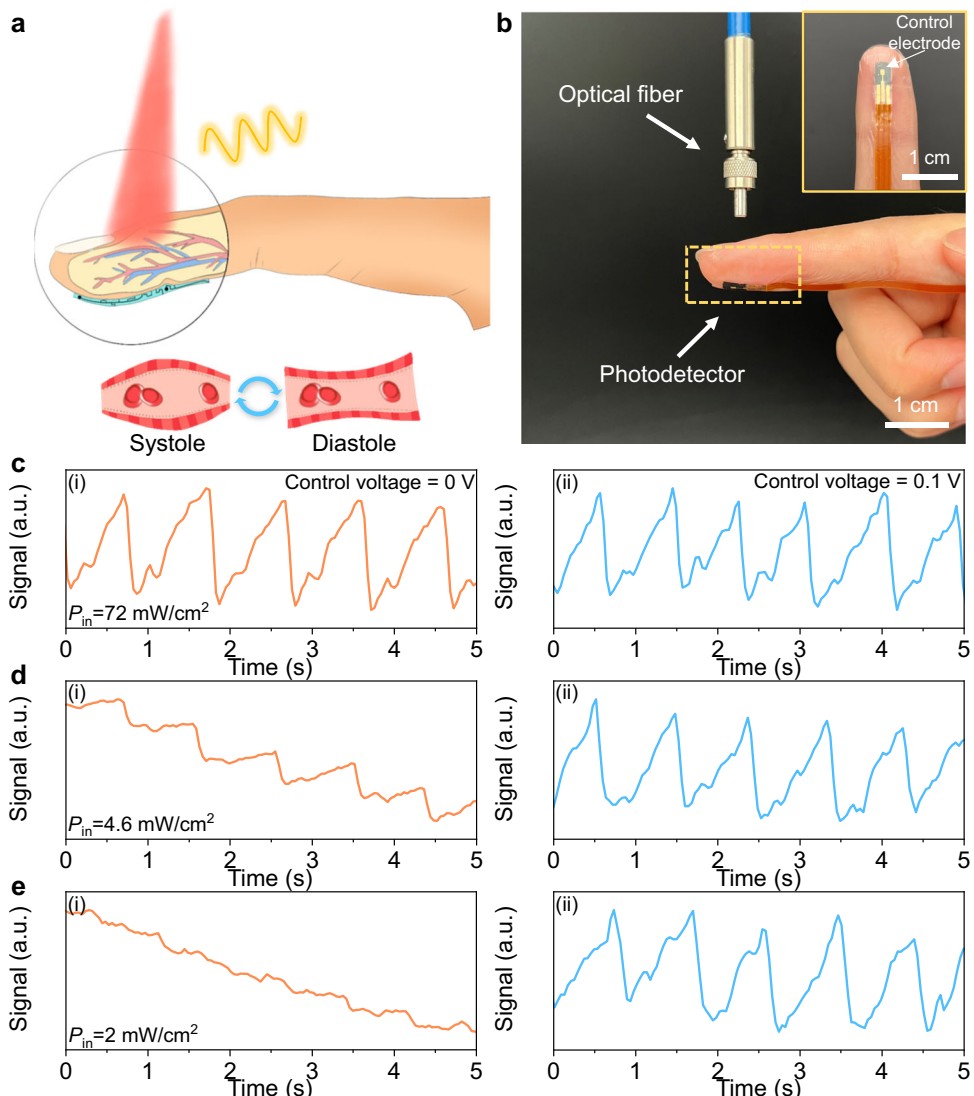

**Fig. 4 | Application of the FPDs for blood pulse signal detection. a** Schematic diagram of the working principle of the PPG test in transmission mode. Volumetric changes in the blood vessels modulate the transmitted light intensity. **b** Photograph of the FPD attached on finger pulp as PPG sensor for recording blood pulse signal. An 800 nm light beam is generated from the optical fiber and reaches the FPD through the finger. Inset shows the top-view photograph of FPD connected with a flexible printed circuit board. **c**–**e** Comparison of PPG signals detected by the FPD under different incident light intensities (72, 4.6, and 2 mW/cm²) when the CE was applied with 0 and 0.1 V. The calculated blood pulse frequency was 67 beats per minute. a.u. arbitrary units.

potential of our method in fabricating low-power consumption wearable electronics.

## Weak light imaging with active-matrix FPD array

Finally, we integrated the perovskite film with a $16 \times 16$ TFT backplane, based on solution-processed $In_2O_3$ semiconductors, to construct the active-matrix FPD (AM-FPD) array (with an active area of $2 \times 2$ cm²). The TFT-based active-matrix FPD array provides merits of high spatial resolution, low signal crosstalk, and low-power consumption[45,46]. A detailed analysis of the dark current and noise of the array is shown in Supplementary Fig. 25. Significantly, to improve its photo-sensing performance, a CE was directly deposited on the perovskite film to apply the modulation electric field. The detailed structure and process scheme of the AM-FPD array, fabricated on the CPI substrate, are shown in Fig. 5a and Supplementary Fig. 26. Photograph of the as-fabricated AM-FPD array wound on a glass rod demonstrating the good mechanical flexibility of the device (Fig. 5b). Each pixel in the array consists of an $In_2O_3$ switching TFT for pixel addressing and a perovskite photoconductor for photo-sensing (Fig. 5c). And the corresponding equivalent circuit and cross-section diagrams of a single

pixel are presented in Fig. 5d and Supplementary Fig. 27, respectively. The statistical distributions of threshold voltage (average $V_{th} = 10.85$ V) and subthreshold swing (average $SS = 0.54$ V dec$^{-1}$) of the 256 ($16 \times 16$) $In_2O_3$ TFTs show a narrow performance distribution, indicating the high-quality imaging capability of the TFT array as a switching backplane (Fig. 5e). The thickness of the SU-8 photoresist is ~5 μm, resulting in a very small capacitance between the control electrode and the $In_2O_3$ semiconductor, so the control voltage (0.1 V) has almost no effect on the mobility of $In_2O_3$ TFT (Supplementary Fig. 28).

Figure 5f shows the real-time relative drain current changes ($\Delta I_D/I_0$) curves when the $In_2O_3$ transistor connected with perovskite film in series at different light intensities (0.5, 1, 2, 5, and 10 μW/cm²). When the control voltage was applied (0.1 V), the real-time current responses showed high stability, low drift, and high SNR value even under weak incident light density, which is critical for high contrast imaging. Still, even under lower working voltage, the specific detectivity of our FPD outperforms documented flexible perovskite PD array, as shown in Fig. 5f and Supplementary Table 1[7,37,47–51].

To evaluate the weak light imaging capability of the AM-FPD array, a weak light beam (520 nm, 500 nW/cm²) was illuminated via a shadow

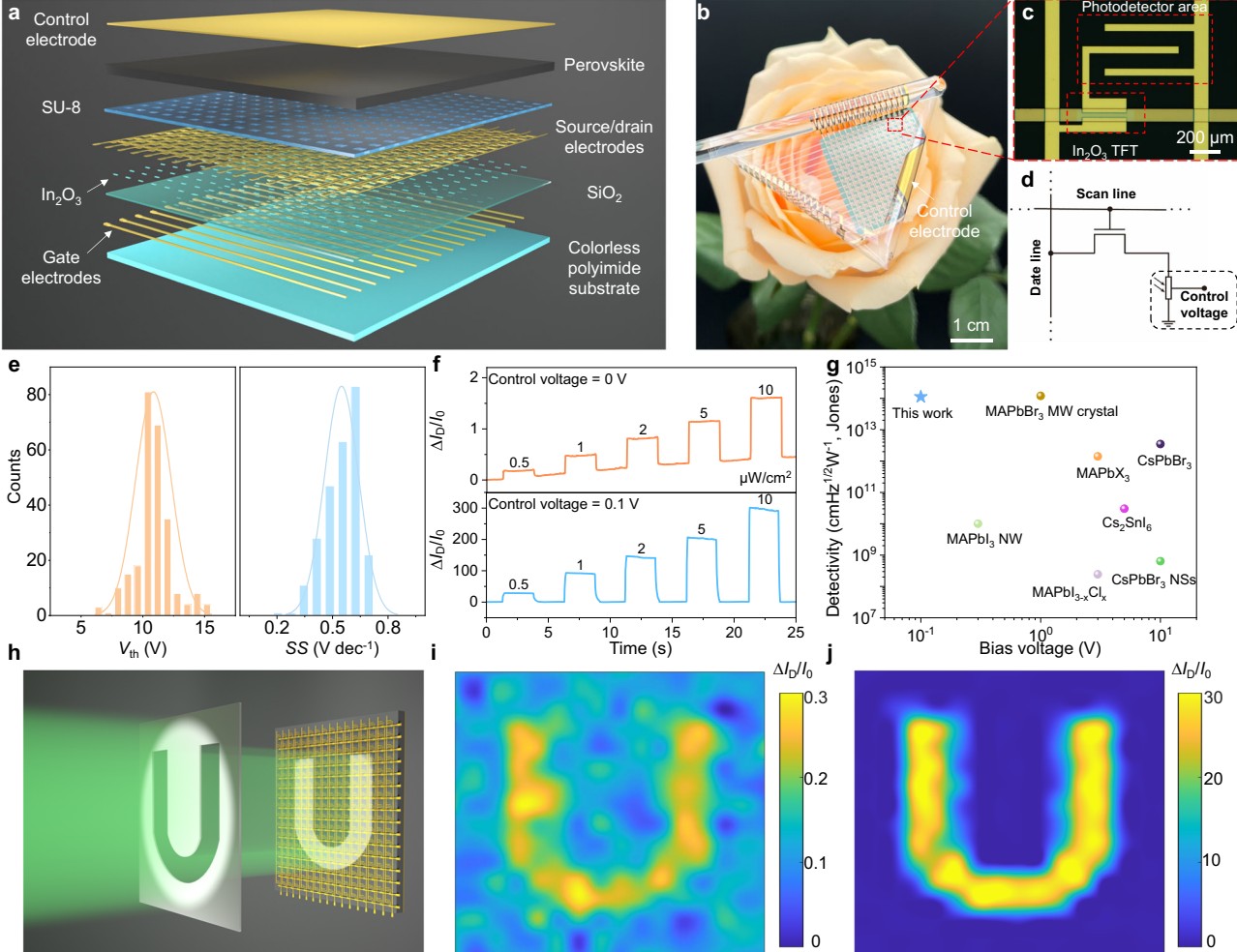

**Fig. 5 | Application of the active-matrix flexible photodetector (AM-FPD) array for weak light imaging. a** Schematic diagram of the 16 × 16 AM-FPD array. **b** Optical image of the AM-FPD array fabricated on a polyimide substrate. **c** The enlarged optical-microscope image of a single pixel in the array, including an $In_2O_3$ transistor and a perovskite photoconductor connected in series. **d** Equivalent circuit diagram of a single pixel. **e** Statistical distributions of the threshold voltage ($V_{th}$) and sub-threshold swing ($SS$) of 256 $In_2O_3$ TFTs. **f** Real-time measurements of normalized drain current changes under various incident light intensities (0.5, 1, 2, 5 and 10 μW/ cm²) of the device at $V_{DS} = 0.1$ V and $V_{GS} = 20$ V when the CE was applied with 0 V (top) and 0.1 V (bottom). **g** Comparison of the detectivity of representative flexible photoconductive-type perovskite photodetector arrays with different operating voltages (NS nanosheet, MW microwire, NW nanowire). **h** Schematic of the light imaging by the AM-FPD array. A hollow 'U' shaped steel mask was placed on the 16 × 16 AM-FPD array for light imaging. **i, j** Normalized current change distribution of the AM-FPD array under 520 nm weak light illumination (500 nW/cm²) when the CE was applied with 0 V (**i**) and 0.1 V (**j**).

mask, with the pattern of character 'U', onto the sensor array, as schematically shown in Fig. 5h. The detailed measurement equipment and process are described in the "Methods". Figure 5i, j present the corresponding relative current changes mappings of AM-FPD without (Fig. 5i) and with (Fig. 5j) applied control voltage. Owing to the large dark current and baseline drift, the patterned shape could not be clearly distinguished by the signal mappings under weak light intensity without control voltage (Fig. 5i). In contrast, high SNR and contrast image of the 'U' pattern was obtained when the control voltage was applied with 0.1 V (Fig. 5j).

## Discussion

In summary, we report an effective strategy to improve the light-sensing performance of photoconductive-type MHPs-based FPD by electrical field modulation. Under the modulation of control voltage, the dark current of MHPs-based FPD can be dramatically reduced. Meanwhile, the photocurrent signal is maintained. As a result, the dark current of FPD was tremendously reduced over 1000 times (from ~5 nA to ~5 pA) and the specific detectivity reached $1.1 \times 10^{14}$ Jones at 520 nm (1 nW/cm²). In addition, ion migration in MHPs was effectively suppressed, and the FPD exhibits a long-term operation capability with

nearly no dark current drift (-1.3 × 10⁻⁵ pA per second) after ~8000 s illumination. Benefitting from the excellent light-sensing performance, the PPG sensor can obtain high-fidelity blood pulse waveform under low incident light intensity (2 mW/cm²), which is critical for the application in low-power consumption wearable electronics. Finally, the MHPs-based photodetector was successfully integrated with a 16 × 16 $In_2O_3$ TFT backplane for high contrast imaging under a weak light environment (500 nW/cm²). This work provides a universal strategy to achieve high-performance MHPs-based FPDs for wearable health monitoring and active-matrix image sensor applications.

## Methods

### Materials

Indium nitrate hydrate (In(NO₃)₃·xH₂O, 99.999%) powder, N,N-dimethylacetamide (DMAC, 99.5%) were purchased from Sigma-Aldrich. 2-methoxyethanol (2-ME, 99.3%), acetylacetone (AcAc, 99%) and ammonium hydroxide (NH₃·H₂O, 28%) were obtained from Alfa Aesar. Formamidinium iodide (FAI, >98.0%), Methylammonium iodide (MAI, >99.0%), Cesium iodide (CsI, >99.0%), Lead (II) iodide (PbI₂, 99.99%), Methylammonium bromide (MABr, >99.0%) and Lead(II) bromide (PbBr₂, 99.99%) were purchased from Tokyo Chemical Industry Co. Ltd

(TCI). Temozolomide (DMSO, 99%), *N,N*-Dimethylformamide (DMF, 98%) and chlorobenzene (CB, 99.8%) were purchased from J&K Scientific. Dichloromethane (DCM) was obtained from Innochem Science & Technology Co., Ltd. Colorless polyimide (CPI) powder was from Zhejiang OCAs New Materials Co. Ltd. Silicon wafer with 100 nm thick $SiO_2$ layer was obtained from Silicon Valley Microelectronics, Inc. Isopropyl alcohol (AR, 99.7%), acetone (AR, 99.5%) and hydrochloric acid (HCl; AR, 36–38%) were provided by China National Medicines Co. Ltd. (Shanghai, China). Poly(perfluoroalkyl vinyl ether) (CYTOP) was purchased from Asahi glass company (Japan) and consisted of CTL-809M (solute) and CT-Solv.180 (solvent). Polymethyl methacrylate (PMMA, 495 K A2) was purchased from Kayaku Advanced Materials.

### Preparation of precursor solution

The precursor solution of $In_2O_3$ was prepared by dissolving 0.3 g $In(NO_3)_3 \cdot xH_2O$ in 10 mL 2-ME. Then, 100 μL AcAc and 35 μL $In(NO_3)_3 \cdot xH_2O$ were added as additives to improve electrical performance[52]. Afterward, the $In_2O_3$ precursor solution was stirred at room temperature overnight. The $FA_{0.92}Cs_{0.04}MA_{0.04}PbI_3$ perovskite precursor (1.4 M) solution was prepared by mixing CsI (29.1 mg) MAI (17.8 mg) FAI (443 mg) $PbI_2$ (1290.83 mg) in 1600 μL DMF and 400 μL DMSO. The $FA_{0.92}Cs_{0.04}MA_{0.04}PbI_3$ perovskite precursor (2.0 M) solution was prepared by mixing CsI (29.1 mg) MAI (17.8 mg) FAI (443 mg) $PbI_2$ (1290.83 mg) in 1120 μL DMF and 280 μL DMSO. The $FA_{0.92}Cs_{0.04}MA_{0.04}PbI_3$ perovskite precursor (2.8 M) solution was prepared by mixing CsI (29.1 mg) MAI (17.8 mg) FAI (443 mg) $PbI_2$ (1290.83 mg) in 600 μL DMF and 400 μL DMSO. The $MAPbI_3$ perovskite precursor (1.4 M) solution was prepared by mixing MAI (445.17 mg) $PbI_2$ (1290.83 mg) in 1600 μL DMF and 400 μL DMSO. The $MAPbBr_3$ perovskite precursor (1.4 M) solution was prepared by mixing MABr (313.52 mg) $PbBr_2$ (1027.63 mg) in 1600 μL DMF and 400 μL DMSO. The precursor solution was stirred at room temperature for 120 min and filtered with a 0.22 μm PTFE filter prior to use. The CPI precursor solution (15 wt%) was prepared by dissolving 1 g CPI powder in 5.7 g DMAC and mixing it in a planetary centrifugal mixer (THINKY ARE-310).

### Crystallization of MAPbBr₃

$PbBr_2$ and MABr (1/1 by molar, 0.4 M) were dissolved in DMF. $MAPbBr_3$ single crystal was grown along with the slow diffusion of the vapor of the antisolvent DCM into the solution.

### Flexible photodetector fabrication

The silicon wafers were cleaned by ultrasonication in acetone, isopropanol and deionized (DI) water sequentially, each for 10 min. Then, the CPI solution was spin-coated on a cleaned silicon wafer at 500 rpm for 60 s and cured on a hot plate (at elevated temperatures of 150, 200, 250 and 300 °C, each for 30 min) in a nitrogen-filled glove box. Next, 50 nm $SiO_2$ was deposited by plasma-enhanced chemical vapor deposition (PECVD; SAMCO Inc. PD-200NL) at 300 °C to serve as a barrier layer. Afterward, the Ni/Au (5/50 nm) electrodes were evaporated by electron beam (e-beam) evaporation (Ei-5z, ULVAC) with shadow masks, forming a channel with width/length (W/L) of 1000/100 μm. 520 nm $FA_{0.92}Cs_{0.04}MA_{0.04}PbI_3$ thin film: $FA_{0.92}Cs_{0.04}MA_{0.04}PbI_3$ perovskite precursor (1.4 M) solution was spin-coated on top of the Au electrodes by a two-consecutive step program at 500 rpm for 10 s and 4000 rpm for 30 s, the antisolvent CB was added at a 5 s countdown. 1 and 2 μm $FA_{0.92}Cs_{0.04}MA_{0.04}PbI_3$ thin film: $FA_{0.92}Cs_{0.04}MA_{0.04}PbI_3$ perovskite precursor (2.0 M for 1 μm and 2.8 M for 2 μm) solution was spin-coated on top of the Au electrodes by a two-consecutive step program at 1000 rpm for 10 s and 4000 rpm for 40 s, the antisolvent CB was added at a 5 s countdown. 520 nm $MAPbI_3$ thin film: $MAPbI_3$ precursor (1.4 M) solution was spin-coated on top of the Au electrodes by one step program at 4000 rpm for 30 s, the antisolvent CB

was added at a 25 s countdown. 520 nm $MAPbBr_3$ thin film: $MAPbBr_3$ precursor (1.4 M) solution was spin-coated on top of the Au electrodes by one step program at 4000 rpm for 30 s, and the antisolvent CB was added at a 15 s countdown. The devices were immediately annealed on a hot plate at 100 °C for 10 min. Then the perovskite films were wiped off with a cotton swab dampened with DMF to form patterns. Finally, the Ni/Au (5/50 nm) electrodes were evaporated on the perovskite films by e-beam evaporation with shadow masks as the CE. The PPG data was acquired from the first author with ethical approval by the ethics committee of Westlake University (20230202ZBW001).

### Flexible photodetector array fabrication

The CPI precursor solution was spin-coated on a cleaned 3-inch silicon wafer and cured on a hot plate by the same process mentioned above. Next, a 50 nm $SiO_2$ barrier layer was formed by PECVD at 300 °C. Photoresist (AR-5350) was spin-coated (at 4000 rpm for 60 s) on the $SiO_2$ film and pre-baked at 105 °C for 4 min. Afterward, the photoresist thin film was exposed to UV light (g-line, 55 mJ/cm²) using a mask aligner (Karl Suss MA6 mask aligner) and removed with the developer (AR 300-26). Ni/Au (5/30 nm) was deposited as bottom gate electrodes by e-beam evaporation on the photoresist and patterned via a lift-off process using acetone, isopropyl alcohol, and deionized water. Subsequently, a 100 nm $SiO_2$ bottom gate insulator was deposited by PECVD at 300 °C. After $O_2$ plasma treatment (PC-300, SAMCO Inc.) at 30 W for 1 min, the prepared $In_2O_3$ precursor solution was spin-coated (3000 rpm, 30 s) on the $SiO_2$ film as the semiconductor layer and pre-baked on a hot plate at 200 °C for 10 min. The AZ-1518 photoresist was spin-coated on the $In_2O_3$ thin film (4000 rpm, 60 s) as an etching mask layer. Afterward, the coated photoresist layer was exposed to UV light (g-line, 450 mJ/cm²) and removed with a developer (AZ 300-MIF). The uncovered $In_2O_3$ thin film was etched with the mixed solution of hydrochloric acid and deionized water (HCl:$H_2O$ = 1:10, v-v) for 10 s. Subsequently, the photoresist was removed and rinsed with acetone, isopropyl alcohol, and deionized water sequentially. Then the patterned $In_2O_3$ thin film was annealed at 300 °C for 1 h. Ni/Au (5/50 nm) was deposited as the source and drain electrodes by e-beam evaporation and patterned by the lift-off process (forming a channel with W/L of 300/30 μm for switch transistor and 1000/100 μm for photodetector) using the same process as for bottom gate electrodes. Subsequently, the SU-8 (2005) photoresist was deposited and patterned via photolithography (i-line, 100 mJ/cm²) to protect the transistor and expose the photo-sensing area. For patterning the perovskite area (2 × 2 cm²), CYTOP (solute: solvent = 1:2) was spin-coated on the device at 5000 rpm for 45 s and patterned by $O_2$ plasma (200 W, 2 min) with a shadow mask. And the $FA_{0.92}Cs_{0.04}MA_{0.04}PbI_3$ perovskite precursor (1.4 M) solution was spin-coated on top of the device by a two-consecutive step program at 500 rpm for 10 s and 4000 rpm for 30 s, the antisolvent CB was added at a 5 s countdown. Then, the device was immediately annealed on a hot plate at 100 °C for 10 min to form perovskite films. Finally, the control Ni/Au (5/100 nm) electrodes were deposited by e-beam evaporation with a shadow mask.

### Material and device characterizations

The crystallization and structural information on the $FA_{0.92}Cs_{0.04}MA_{0.04}PbI_3$, $MAPbI_3$ and $MAPbBr_3$ films were obtained by X-ray diffraction (XRD, D8 Advance, Bruker). The SEM images were obtained by the field emission scanning electron microscope (FESEM, Zeiss Gemini 500). The root-mean-square (RMS) surface roughness of the perovskite films was measured by atomic force microscopy (AFM, Bruker Dimension ICON). The photoluminescence (PL) was detected by the confocal microscope spectrometer (Alpha300, WITec). The 532 nm excitation light was generated by Pico Quant PDL 800-D. UV-visible (UV-vis) absorption spectra were measured using a JASCO V-770 spectrophotometer. The optical micrographs of the devices were

obtained by digital microscope (Olympus DSX1000). Ultraviolet photoelectron spectroscopy (UPS) measurements were performed by an ESCALAB 250Xi X-ray Photoelectron Spectroscopy (Thermo Fisher Scientific, USA). UPS employed the He I (21.22 eV) as the excitation source with an energy resolution of 50 meV.

The electrical characteristics of halide perovskite photoconductor and $In_2O_3$ transistor were measured by using a source meter (Keithley 2636B) and a semiconductor parameter analyzer (4200A-SCS, Keithley) integrated with a probe station system in a shielded dark box at room temperature. The optical stimuli were generated by a 405/520/600/800 nm semiconductor laser (Changchun New Industries Optoelectronics Tech Co., Ltd, China) and modulated by a Tektronix AFG3152C function generator. The stable optical pulse illumination is modulated by a mechanical chopper (SSI-instrument, OE3001). The light intensity was measured using a standard silicon photodiode (S120VC, Thorlabs). For imaging of spatial light distribution, three PXI source measure units (NI PXIe-4138) and a PXI matrix switch module (NI PXIe-2531) were assembled in a PXI Chassis (NI PXIe-1073) to construct a data acquisition system. The source, drain, gate and control electrodes of the flexible photodetector array were connected to customized flexible cables using anisotropic conducting paste. And the image-sensing characteristics of the flexible photodetector array were measured using the light stencil projection method. A 2 mm thick stainless steel mask plate with a hollowed-out figure of 'U' was placed on the $16 \times 16$ flexible photodetector array. Subsequently, the signals of individual pixels were measured in sequence under a 520 nm light projection. The imaging data detected by the flexible photodetector array was processed and linearly interpolated by MATLAB R2021a.

The subthreshold swing values (*SS*) of the transistors were extracted with:

$$SS = \frac{dV_{GS}}{d\log_{10}(I_D)} \tag{4}$$

Where *W* and *L* are the channel width and length of the transistors, respectively. $I_D$ and $V_{GS}$ represent the drain current and gate voltage, respectively. $C_{ox}$ is the capacitance of the gate dielectric layer.

## Data availability
The data that support the plots within this paper and other findings of this study are available from the corresponding author upon request. Source data are provided with this paper.

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

## Acknowledgements

This work was supported by the National Natural Science Foundation of China (Grant No. 62174138, 62074136, 52273307), Westlake Multi-disciplinary Research Initiative Centre (MRIC) Seed Fund (Grant No. MRIC20200101), Leading Innovative and Entrepreneur Team Introduc-tion Program of Zhejiang (Grant No. 2020R01005) and Natural Science Foundation of Zhejiang province of China (LZ23F050005). We thank Westlake Centre for Micro/Nano Fabrication, the Instrumentation and Service Centre for Physical Sciences (ISCPS), and the Instrumentation and Service Centre for Molecular Sciences (ISCMS) at Westlake Uni-versity for the facility support and technical assistance.

## Author contributions

B.Z. and Y.(M.)Y. conceived the idea and supervised the overall project. Y.T. and P.J. conducted most of the experiments. Y.T. analyzed the data and made the figures with the assistance of P.J. Y.W., D.L., Y.C., P.R., W.F., K.L., H.R., X.X., R.W., assisted with material characterizations, device fabrications or data analysis. W.F. fabricated the $MAPbBr_3$ single crystal. Y.T. and B.Z. wrote the manuscript. All authors contributed to the review of the manuscript.

## Competing interests

The authors declare no competing interests.
