## [Peer Review file · Nature Communications]

Enabling low-drift flexible perovskite photodetectors by electrical modulation for wearable health monitoring and weak light imagingREVIEWER COMMENTS

Reviewer #1 (Remarks to the Author):

This paper reports the results of the application of the field modulation method to metal halide perovskite (MHP), finding that the dark current of flexible photodetectors (FPDs) can be reduced by more than 1000 times (from ~ 5 nA to ~ 5 pA). This structure significantly reduces ion migration, and the proposed FPD achieves long-term continuous operating stability (~ 8000 s), low signal drift ($\sim 4.2 \times 10^{-4}$ pA/sec), and ultra-low dark current drift ($\sim 1.3 \times 10^{-5}$ pA/sec). These characteristics of proposed FPD are very excellent. Furthermore, the usefulness of the system as an active-matrix FPD configuration has been demonstrated by constructing a FPD active array and actually acquiring images. So far, FPDs have been vigorously researched and developed by many groups. However, MHPs are affected by defects and ion migration in polycrystalline materials, and the associated large dark currents have been an issue. Furthermore, in continuous use, the electrical characteristics often became unstable, which led to deterioration of detection limits and significant challenges in long-term operation.

This paper is an effort to overcome this issue by employing a structure that allows field effects to be applied within the MHP. The strategy of electrical property modulation by integrating field-effect transistor structures and FPDs, the high signal-to-noise ratio (SNR) characteristics, and the successful development of photo-sensors for wearable photopotentiometer (PPG) weak light imaging and active-matrix photodetector arrays are outstanding.

However, I believe that the following perspectives need to be more clearly described and demonstrated.

1:

I think the method of controlling dark current by field effect with three electrodes, similar to a transistor, is superior from the viewpoint of current and electric field control. On the other hand, it is necessary to place the top electrodes and furthermore, to perform wiring to the three electrodes. Such complicated electrodes and wirings reduce the area of the "aperture" that can receive light. The presence of electrodes is a major issue when one wants to obtain higher resolution images. What changes in characteristics will occur if the ratio of the electrode area to the light-receiving area that can receive light is changed? Please describe how the highest value shown in this paper, " 1.1×10^{14} Jones at 520 nm wavelength," changes when the area ratio of the electrode portion and the light-receiving portion is changed.

2:

The discussion of the contact resistance at the interface between the electrode and the MHP surface, and the injection barrier associated with the difference in work function is important, but not much is mentioned in this paper. What results would be obtained if other electrodes were used, such as those with different work functions or transparency?

3:

Baseline drift is clearly visible in many experimental results, but please provide a more detailed description of its origin. Also, baseline drift is often a transient characteristic, which means that the slope may change over time. This study does not show a relationship between this drift and time. I would like to see experimental results and more detailed discussion regarding the drift, such as whether it is a thermal effect or the effect of crystal defect trapping.

4:

The dark current in the proposed FPD can be suppressed by the electric field effect, but the crosstalk and weak leakage current in the wiring and FPD with three electrodes will occur when the active matrix structure shown in Figure 5 is used. The higher the resolution, the more leakage current associated with crosstalk at wirings and/or electrodes is expected to increase.

In the case of readout using such an active matrix structure, the total amount of dark current as a system is expected to be larger, mainly due to the crosstalk and leakage current of the wiring, not the dark current of each individual optical sensor. Please provide experimental results and discuss the dark current as a system. The experimental results (absolute value of dark current as a system) are not shown in Figure 5 and other experiments, but they are very important for applications.

Reviewer #2 (Remarks to the Author):

The manuscript of Tang and co-workers reports flexible perovskite photodetectors with ultra-low dark current. The authors described a novel device architecture comprising an addition of a control electrode in a photoconductor configuration, which controls dark current. This mechanism is well justified by simulation and data in the manuscript and, therefore, very convincing. Finally, the authors proved the concept with flexible substrates for wearable bio-signal detection and in an imager architecture. These real-world applications make the manuscript very appealing for the perovskite community and, more in general, to research groups in the semiconductor field. I have some comments that should be considered before accepting the manuscript for publication in Nature Communications.

- 1) The advantage of this structure should be better described compared to for example a TFT configuration.
- 2) Hysteresis data must be reported, I am expecting that without selective contacts, the hysteresis is quite high for this configuration.
- 3) Why has the same contact (Au) been used for the control and ground and signal electrodes? What would happen in case of different metal contact?
- 4) A cross-section SEM will help the description of the configuration.
- 5) Similarly, SEM will be beneficial to appreciate the quality of the perovskite layers coated on top of the control electrode.

Reviewer #3 (Remarks to the Author):

Photodetectors can directly convert optical signals into electrical signals, and play an important role in automatic driving, environmental monitoring, medical imaging, and military fields. Perovskite material has excellent photoelectric performance of direct band gap, large absorption coefficient and electron hole diffusion length, which opens a new door for the research of photodetectors. In this paper, The authors propose a significant electric field modulation strategy, based on which the dark current is significantly reduced and the ion migration is effectively suppressed. However, there are still problems in mechanism innovation and conclusion verification. The details are as follows:

1. In a recent paper by the author, a similar control method was used, that is, the introduction of additional control electrodes to suppress the dark current of the X-ray detector. Compared with this manuscript, what are the innovations in physical mechanism and technical methods?
2. The author introduces a third electrode on the other side of the photoconductor to suppress the dark current. The introduced third electrode is similar to the gate in the transistor, and What is the difference between both them?
3. The author mentioned that the iodine ion migration was inhibited by introducing the control electrode, but no direct evidence was given to prove this.
4. Is the response time related to the control electrode voltage?
5. What is the carrier mobility of the device in the thin film transistor array? Compared with similar devices, does the carrier mobility increase due to the introduction of the third electrode?
6. In Fig. 3h and Fig. 5f, the reason why the photocurrent rises first and then falls?

7. The author shows a response pulse of photoconductivity under extremely weak light intensity in the supporting information Fig. 10. It is suggested that the author increase at least 10 stable response periods to increase credibility.

8. Influence of film quality on voltage selection of control electrode needs to be pointed out.

Responses to NCOMMS-23-06143

We appreciate all reviewers' constructive comments and valuable suggestions. We have carefully read through the reviewers' comments and thoroughly addressed all the requests and concerns. And hopefully, our responses can release your concerns.

Following are our point-by-point responses:

REVIEWER COMMENTS

Reviewer #1 (Remarks to the Author):

This paper reports the results of the application of the field modulation method to metal halide perovskite (MHP), finding that the dark current of flexible photodetectors (FPDs) can be reduced by more than 1000 times (from ~ 5 nA to ~ 5 pA). This structure significantly reduces ion migration, and the proposed FPD achieves long-term continuous operating stability (~ 8000 s), low signal drift ($\sim 4.2 \times 10^{-4}$ pA/sec), and ultra-low dark current drift ($\sim 1.3 \times 10^{-5}$ pA/sec). These characteristics of proposed FPD are very excellent. Furthermore, the usefulness of the system as an active-matrix FPD configuration has been demonstrated by constructing a FPD active array and actually acquiring images. So far, FPDs have been vigorously researched and developed by many groups. However, MHPs are affected by defects and ion migration in polycrystalline materials, and the associated large dark currents have been an issue. Furthermore, in continuous use, the electrical characteristics often became unstable, which led to deterioration of detection limits and significant challenges in long-term operation.

This paper is an effort to overcome this issue by employing a structure that allows field effects to be applied within the MHP. The strategy of electrical property modulation by integrating field-effect transistor structures and FPDs, the high signal-to-noise ratio (SNR) characteristics, and the successful development of photo-sensors for wearable photopotentiometer (PPG) weak light imaging and active-matrix photodetector arrays are outstanding.

However, I believe that the following perspectives need to be more clearly described and demonstrated.

Response: Thank you very much for your positive comments and constructive suggestions. We have carefully addressed the issues and we have revised the manuscript accordingly.

Q1: I think the method of controlling dark current by field effect with three electrodes, similar to a transistor, is superior from the viewpoint of current and electric field control. On the other hand, it is necessary to place the top electrodes and furthermore, to perform wiring to the three electrodes. Such complicated electrodes and wirings reduce the area of the "aperture" that can receive light. The presence of electrodes is a major issue when one wants to obtain higher resolution images. What changes in characteristics will occur if the ratio of the electrode area to the light-receiving area that can receive light is changed? Please describe how the highest value shown in this paper, " 1.1×10^{14} Jones at 520 nm wavelength," changes when the area ratio of the electrode portion and the light-receiving portion is changed.

Response: Thanks for the questions. In our work, the common electrode (top electrode, Fig. R1, Supplementary Figure 27) was applied to provide electric field control, which doesn't block light absorption, as the light reaches the device through bottom side (penetrating colorless polyimide and SiO₂ layers). Thus, the common electrode does not influence the pixel density in the array.

Fig. R1 | Cross-sectional of a single pixel of AM-PFD array.

On the other hand, the employment of control electrodes—defined and patterned by photolithography—did reduce the area of the “aperture”. To obtain higher-resolution images, the footprints of these electrodes need further downscaling.

Following the suggestion and question, we fabricated the FPDs with different perovskite channel lengths (100, 50, and 25 μm), and the electrodes were fixed at 1000 μm in length and 100 μm in width (with an area of $2 \times 10^{-3} \text{ cm}^2$). And the ratio of the electrode area ($A_{\text{electrode}}$) to the light-receiving area ($A_{\text{light-receiving}}$) is 2, 4, 8, respectively. As shown in Fig. R2a, the red dashed area represents the light receiving area ($A_{\text{light-receiving}}$), and the blue dashed area represents the area of the whole device (A_{device}).

Fig. R2 | Comparison of the optoelectronic properties of FPD with different ratios of electrode area to the light-receiving area. a,b, Sketch image and photocurrent responses of FPDs of FPDs with different channel lengths (100, 50, and 25 μm). **c,** Noise power density of the dark current of the electrical field modulated FPDs. **d,** The calculated specific detectivity according to the device areas and the light-receiving areas.

A 520 nm light with a power intensity of 1 nW/cm^2 was used to investigate the photoelectric properties of the FPDs with different areas. Fig. R2b shows the photocurrent of the FPDs with different channel lengths when the working voltage and control voltage were set to 0.1 V.

Despite the differences in light-receiving area, the photocurrent values of the FPDs maintained similar value at ~8 pA, because the reduction of the channel length increases the electric field and reduces the transit time of charge carriers [*ACS Photonics* **5**, 4111-4116 (2018)].

According to Ohm's law and the definition of conductivity (σ), we can derive the expression for dark current density (J),

$$J = \sigma \frac{V}{L}$$

where V is the bias voltage, L is the channel length. In general, in conventional photoconductive-type photodetectors, the dark current will increase as the channel length decreases because the conductivity of the photoactive material does not change. However, through our electric field modulation strategy, the dark current can be effectively suppressed regardless of channel length. Fig. R2c shows the noise power density of the dark current of FPDs with different channel lengths. The specific detectivity (D^*) of FPDs with different channel lengths can be evaluated by the following equation:

$$D^* = \frac{R\sqrt{AB}}{I_{noise}}$$

Where R is the responsivity, B is the bandwidth, and I_{noise} is the measured total noise.

To extract D^* in the lateral photoconductive-type devices (Fig. R2d, blue dots), A usually represents the active area which corresponds to the $A_{light-receiving}$ [*Nat. Photon.* **13**, 1-4 (2019)]. If the entire device area (A_{device}) is used as $A_{light-receiving}$ to extract the specific detectivity (D^*_{ref} , Fig. R2d, orange dots), the value of the calculated D^*_{ref} will be slightly reduced (compared with D^*) due to the influence of the electrode area.

Still, although with some fluctuations, both D^* and D^*_{ref} maintained at high values with different ratios of electrode area to the light-receiving area, as shown in Fig. R2d. This further illustrates the effectiveness of our method. Detailed parameters of FPDs with different channel lengths are shown in Table R1.

Table R1 | The detailed parameters of FPDs with different channel lengths and ratios of electrode area to the light-receiving area.

Channel Length	I_{noise} (A Hz ^{-1/2})	$\frac{A_{\text{electrode}}}{A_{\text{light-receiving}}}$	$A_{\text{light-receiving}}$ (cm ²)	A_{device} (cm ²)	D^* (Jones)	D^*_{ref} (Jones)
25 μm	3.27×10^{-15}	8	2.5×10^{-4}	2.25×10^{-3}	1.56×10^{14}	5.22×10^{13}
50 μm	5.67×10^{-15}	4	5×10^{-4}	2.5×10^{-3}	6.36×10^{13}	2.84×10^{13}
100 μm	2.25×10^{-15}	2	1×10^{-3}	3×10^{-3}	1.12×10^{14}	6.47×10^{13}

Q2: The discussion of the contact resistance at the interface between the electrode and the MHP surface, and the injection barrier associated with the difference in work function is important, but not much is mentioned in this paper. What results would be obtained if other electrodes were used, such as those with different work functions or transparency?

Response: We thank the reviewer for providing this thoughtful comment. The work function of metal electrodes and the band structure of perovskite is very important for charge carrier injection and transport.

First, the bandgap of perovskite can be extracted to be 1.54 eV from the PL spectra by:

$$E_g = \frac{1240}{\lambda_p}$$

Where λ_p is the peak wavelength (803 nm) of PL. Then, we utilized ultraviolet photoelectron spectroscopy (UPS) to define the Fermi level and the valence band maximum (VBM) of perovskite (Fig. R3). UPS measurements were performed by ESCALAB 250Xi X-ray Photoelectron Spectroscopy (Thermo Fisher Scientific, USA) with the He I (21.22 eV) as the excitation source. The Fermi level, VBM and conduction band minimum (CBM) of perovskite were extracted at -5.12, -5.75, and -4.21 eV, respectively.

Fig. R3 | Ultra-violet photoelectron spectroscopy (UPS) of perovskite. a, The secondary electron cutoff region **b,** and the valence band onset of UPS of perovskite film.

Fig. R4 | Energy band diagrams for the Au/perovskite and Ag/perovskite interface.

When two metals with different work functions, Au (-5.1 eV) and Ag (-4.2 eV) [*Adv. Sci.* **9**, 2203683 (2022)], form contacts with perovskite, different injection barriers could be expected. As shown in the energy band diagram (Fig. R4), perovskite materials forms Schottky contacts with Ag, but could form Ohmic contact with Au. Thus, Au/perovskite contact outperforms Ag/perovskite contact in carrier transport, as shown in Fig. R5a. Meanwhile, Ag contact is less stable than Au and it may react with perovskite.

In addition, we fabricated a device with asymmetric electrodes (Ag/perovskite/Au), whose current-voltage curve deviated from the origin (Fig. R5b), which also indicated the existence of the Schottky barrier [*Adv. Optical Mater.* **10**, 2100786 (2022)]. Also, Fig. R5c shows that under the same 1 mW/cm² light illumination, the Au/perovskite/Au device exhibited the highest photocurrent.

Fig. R5 | Characteristics of the FPD fabricated by different electrodes. **a**, Current-voltage characteristics of Au/perovskite/Au and Ag/perovskite/Ag FPDs. **b**, Current-voltage characteristics of Ag/perovskite/Au FPD. **c**, Photocurrent of FPD with different electrodes under 1 mW/cm² illumination.

In addition, we also fabricated devices with transparent ITO electrodes. Fig. R6 a and b show the light transmittance spectra of Au and ITO electrodes, respectively. The Au electrode is almost opaque in the visible light range, while the light transmittance of ITO exceeds 80%. Thus, using ITO as the electrode could potentially improve the photoelectric response of the FPD (increase light absorption area).

Fig. R6 | The effect of electrode transparency on FPD. **a,b**, Optical transmittance spectrum of Au electrode (50 nm) and ITO electrode (50 nm). **c**, Photocurrent response of the FPDs fabricated by ITO electrodes under 520 nm light (1 Hz, 0.1 mW/cm²) illumination.

However, the magnetron sputtering (RF) deposition of ITO electrodes could deteriorate the perovskite film and leads to device performance degradation. As shown in Fig. R6c, the FPD device with ITO electrodes showed very high dark current and serious current drift, resulting in poor photoelectric detection performance. In the future, optimized deposition methods and fabrication processes could be endeavored to avoid perovskite film damage and thus enabling better photodetection performance with transparent electrodes.

The corresponding results and discussions have been incorporated into the revised manuscript (page 12, paragraph 1) and supplementary materials (Figs. S21-23).

Q3: Baseline drift is clearly visible in many experimental results, but please provide a more detailed description of its origin. Also, baseline drift is often a transient characteristic, which means that the slope may change over time. This study does not show a relationship between this drift and time. I would like to see experimental results and more detailed discussion regarding the drift, such as whether it is a thermal effect or the effect of crystal defect trapping.

Response: Thanks for the instructive comment. We believe that ion migration is the main cause of baseline drift. Due to the weak ionic-bonding nature of perovskites, ions tend to migrate under electrical bias, which leads to ionic conductivity and further causes baseline drift [Nat. Commun. **6**, 7497 (2015)]. In order to better illustrate the relationship between baseline drift and time, we extracted the dark current variation with time in Fig. 3g,h, as shown in Fig. R7a,b. By control voltage modulation, not only the value of the dark current is reduced, but also the ion migration near the signal electrode is suppressed, thus resulting a more stable baseline (Fig. R7b).

Furthermore, as indicated by the reviewer's suggestion, ion migration is directly related to temperature [Adv. Energy Mater. **6**, 1501803 (2016)] and crystal defects [Energy Environ. Sci. **9**, 1752-1759 (2016)]. The relationship between the ion migration rate (r_m) and temperature (T) can be expressed by the following expression:

$$r_m \propto \exp\left[-\frac{E_A}{k_B T}\right]$$

Where E_A is activation energy and k_B is Boltzmann's constant.

Following the suggestion on the studying the effect of temperature on ion migration, we measured the current-time curves of FPD at different temperatures (from -30 to 50 °C). As shown in Fig. R7c, as the test temperature increases, the dark current increases and the baseline drift becomes more obvious. In addition, ion migration in polycrystalline films mostly occurs

via point defects or grain boundaries, due to the smaller ion activation energy at defect sites [Nat. Commun. **10**, 1989 (2019)].

On the other hand, to study the effect of crystal defect trapping on ion migration, we fabricated devices based on MAPbBr₃ single crystal perovskite. As shown in Fig. R7d, due to its lower defect density, the baseline current value of single crystal perovskite is very low (~0.3 nA, an order of magnitude lower than that of polycrystalline perovskite), and no significant drift was observed.

Therefore, both crystal defects and temperature could contribute to the ion migration and thus lead to baseline drift. Fig. R7 has been added as Supplementary Figure 16 to the revised version.

Fig. R7 | The origin of FPD’s baseline drift. **a,b** The relationship between the baseline drift and time extracted from Figure 3g,h. **c**, Current-time curves of baseline drift at different temperatures. **d**, Current-time curve of baseline drift of MAPbBr₃ single crystal perovskite.

Q4: The dark current in the proposed FPD can be suppressed by the electric field effect, but the crosstalk and weak leakage current in the wiring and FPD with three electrodes will occur when the active matrix structure shown in Figure 5 is used. The higher the resolution, the more

leakage current associated with crosstalk at wirings and/or electrodes is expected to increase. In the case of readout using such an active matrix structure, the total amount of dark current as a system is expected to be larger, mainly due to the crosstalk and leakage current of the wiring, not the dark current of each individual optical sensor. Please provide experimental results and discuss the dark current as a system. The experimental results (absolute value of dark current as a system) are not shown in Figure 5 and other experiments, but they are very important for applications.

Response: We thank the reviewer's important comment. In order to obtain the signals of the array sensors, there are three widely used wiring methods: direct addressing, passive addressing, and active addressing, as shown in Fig. R8 [Adv. Intell. Syst. 4, 2100253 (2022)]. When the number of sensing elements increases, the interconnects between the pixels will be increased dramatically with the direct addressing method, which hinders the practical application [ACS Appl. Mater. Interfaces 11, 29014-29021 (2019)]. The passive addressing with crossbar structure is simple from the design and fabrication perspective but suffers from large crosstalk, slow rate, and high-power consumption [Nano-Micro Lett. 13, 140 (2021)]. However, within the active-matrix sensing array system, thin film transistors (TFTs) act as switching elements to select individual pixels for signal readout. Hence, the TFTs-based active matrix addressing approach can effectively reduce signal crosstalk and power consumption. Furthermore, the active-matrix backplane composed of TFTs array can provide a universal platform for sensing elements integration with high pixel density [Nature 555, 83-88 (2018)].

Fig. R8 | Digital optical images of **a**, the direct addressing sensor array [ACS Appl. Mater. Interfaces 11, 29014-29021 (2019)], **b**, passive matrix addressing sensor arrays [Nano-Micro Lett. 13, 140 (2021)] and **c**, the active matrix addressing sensor array [Nature 555, 83-88 (2018)].

Therefore, we use a transistor in series with a perovskite photodetector to construct an active matrix FPD (AM-FPD) array, and the corresponding equivalent circuit diagram is shown in Fig. R9a. In such an array, each photodetector pixel is controlled and addressed by a TFT, providing arbitrary selecting, and avoiding signal crosstalk. During the testing process, the corresponding row and column electrodes are scanned at a certain frequency (similar to the refreshing rate of a cellphone display) and selected through matrix switch module (NI PXIe-2531) to turn on a specific transistor, thereby reading the perovskite photodetector signal.

Fig. R9 | Detailed analysis of the dark current and noise of the array. **a**, Circuit diagram of the sensor array. **b**, Take $R_{16,16}$ as an example to analyze the addressing process and the origin of dark current. The gate voltage ($V_{GS}=20$ V) and drain voltage ($V_{DS}=0.1$ V) are respectively applied to the sixteenth row (orange line) and sixteenth column (blue line) to read the signal of $R_{16,16}$. **c**, A simplified circuit of the case of the addressed single pixel. **d**, Current-time curve of a single pixel actually measured in the array.

Take photodetector pixel $R_{16,16}$ in the array as an example, we analyze the addressing process and the origin of dark current in the following. The gate voltage ($V_{GS}=20$ V) and drain voltage ($V_{DS}=0.1$ V) are respectively applied to the 16th row (orange line) and 16th column (blue line)

to read the signal of $R_{16,16}$ (Fig. R9b). Since other transistors are in the off state (the gate voltage is not applied), the current in the circuit is almost all contributed by $R_{16,16}$ at this time. The simplified equivalent circuit diagram of the addressed single pixel is shown in Fig. R9ci. The transistor is turned on at this time, so the transistor can be equivalent to a resistor (R_{TFT}), whose resistance value is much smaller than the perovskite photoconductor ($R_{TFT} \ll R_{16,16}$). As shown in Fig. R9cii and R8ciii, when the control voltage is applied, the dark current (blue arrow, $I_{\text{Dark current}}$) measured by the instrument is shunted by the control electrode, so the measured $I_{\text{Dark current}}$ value can be significantly reduced.

In the actual test, the absolute value of the measured dark current can also be reduced to ~ 5 pA, but its random noise is significantly higher than the current signal measured directly on the perovskite, which is mainly due to the environmental noise and transistor device noise (Fig. R9d). In future applications, the detection performance of the array can be improved by improving the quality of addressing transistors, optimizing the circuit wiring structure, and enhancing the electromagnetic shielding ability of the device to obtain higher-quality current data. The corresponding discussion and data have been supplemented in the revised manuscript (page 14, line 320) and supplementary material (Fig. S25).

Reviewer #2 (Remarks to the Author):

The manuscript of Tang and co-workers reports flexible perovskite photodetectors with ultra-low dark current. The authors described a novel device architecture comprising an addition of a control electrode in a photoconductor configuration, which controls dark current. This mechanism is well justified by simulation and data in the manuscript and, therefore, very convincing. Finally, the authors proved the concept with flexible substrates for wearable bio-signal detection and in an imager architecture. These real-world applications make the manuscript very appealing for the perovskite community and, more in general, to research groups in the semiconductor field. I have some comments that should be considered before accepting the manuscript for publication in Nature Communications.

Response: We would like to gratefully thank the reviewer for your important comments and for your evaluation. The manuscript has been revised with full consideration of your comments.

Q1: *The advantage of this structure should be better described compared to for example a TFT configuration.*

Response: Thanks for your kind suggestion. The typical configuration of a bottom gate top contact (BGTC) transistor is shown in Fig. R10. Compared with transistors configuration, our proposed structure has no dielectric layer and the control electrode is in direct contact with the semiconductor material. The control electrode in our structure can directly participate in the current transport process between the source and drain electrodes, so it has a stronger control effect on the dark current, and the value of the dark current can be reduced to zero or even a negative value (Fig. 2c). Therefore, our structure can achieve lower dark current values compared to a TFT configuration. Corresponding comments have been added to the revised manuscript (page 3, line 70) and supplementary material (Fig. S1).

Fig. R10 | Schematic of a bottom gate top contact (BGTC) transistor.

Q2: Hysteresis data must be reported, I am expecting that without selective contacts, the hysteresis is quite high for this configuration.

Response: We thank the reviewer for the insightful comments. Previous reports have demonstrated that mobile ions and trapping defects inside perovskite films are the main causes of hysteresis [*Adv. Mater.* **31**, 1805214 (2019)]. As expected, there is hysteresis in the forward and reverse current-voltage scan curves of FPD with Au electrodes (Fig. R11a). In addition to ion migration, direct chemical reactions between electrodes and metals can also lead to severe hysteresis [*ACS Energy Lett.* **1**, 595-602 (2016)]. The current-voltage curves of the devices fabricated with aluminum (Al) electrodes exhibited serious hysteresis. Therefore, the Au electrode is a relatively stable metal electrode in contact with the perovskite. In future work, the hysteresis can be further reduced by introducing an interfacial modification layer between the perovskite and the electrode, and developing new stable electrode materials [*Adv. Sci.* **9**, 2203683 (2022)]. The corresponding discussion and data have been added to the revised manuscript (page 12, line 268) and supplementary material (Fig. S24).

Fig. R11 | Forward and reverse current-voltage scan curves of FPD with Au and Al electrodes at the dark condition.

Q3: Why has the same contact (Au) been used for the control and ground and signal electrodes? What would happen in case of different metal contact?

Response: We thank the reviewer for this comment. The reason we use the Au electrode is that it has better chemical stability than other electrodes, and its work function matches that of

perovskite, so it can have a better carrier transport effect [*Adv. Sci.* **9**, 2203683 (2022)]. When two metals with different work functions, Au (-5.1 eV) and Ag (-4.2 eV) [*Adv. Sci.* **9**, 2203683 (2022)], are used as electrodes to fabricate FPD, the energy band diagram of the semiconductor-metal interface is shown in Fig. R12. According to the energy band bending theory, there exists an obvious Schottky barrier between perovskite and Ag electrode, which hinders the transport of charge carrier [*Chem. Rev.* **112**, 5520-5551 (2012)]. However, since the work function of the Au electrode (-5.1 eV) is close to the Fermi level of the perovskite (-5.12 eV), there is almost no barrier at the Au/perovskite interface, which facilitates the carrier transport.

Fig. R12 | Energy band diagrams for the Au/perovskite and Ag/perovskite interface.

As shown in the current-voltage characteristics (Fig. R13a), the contact resistance with Ag electrode is much larger than that of Au electrodes, which is caused by the high Schottky barrier at the Ag electrode/perovskite interface. In addition, we fabricated a device with asymmetric electrodes (Ag/perovskite/Au), whose current-voltage curve deviated from the origin, which also indicated the existence of the Schottky barrier.

Fig. R13c shows that under $1\text{mW}/\text{cm}^2$ illumination, device with both Au electrodes exhibited the highest photocurrent than symmetric Ag-Ag or asymmetric Ag-Au electrodes. Therefore, we prefer to choose Au electrodes with work function matching and chemical stability to fabricate devices for better optoelectronic performance. The corresponding results and discussions have been incorporated into the revised manuscript (page 12, paragraph 1) and supplementary materials (Figs. S22 and S23).

Fig. R13 | Characteristics of the FPD fabricated by different electrodes. a, Current-voltage characteristics of Au/perovskite/Au and Ag/perovskite/Ag FPDs. **b,** Current-voltage characteristics of Ag/perovskite/Au FPD. **c,** Photocurrent of FPD with different electrodes under 1 mW/cm^2 illumination.

Q4: A cross-section SEM will help the description of the configuration.

Response: Thank you for the valuable comment. Because the perovskite film thickness is only 500 nm and the channel length is $100\ \mu\text{m}$, it is difficult to show the specific details of the device in a low-magnification scanning electron microscopy (SEM) image (Fig. R14a). SEM image showing detailed device structure is provided in Fig. R15 in the next response. We supplemented a detailed 3D diagram (Fig. R14b) and added it to our revised manuscript (page 7, line 150) and supplementary materials (Fig. S4) to describe the feature size of the device, where the channel length is $100\ \mu\text{m}$, the channel width is $1000\ \mu\text{m}$, and the thickness of the perovskite film is 500 nm.

Fig. R14 | The configuration of FPD. a, Cross-sectional SEM image of the FPD. **b,** A 3D image of the detailed feature dimensions of the electrical field modulated FPD.

Q5: Similarly, SEM will be beneficial to appreciate the quality of the perovskite layers coated on top of the control electrode.

Response: We thank the reviewer for the valuable comment. To illustrate this point, we fabricated an electrical field modulated photodetector on the silicon substrate. The cross-sectional SEM image is shown in Fig. R15, and there is intimate contact between the perovskite and the Au control electrode without observable voids, implying good quality of the perovskite film. The corresponding results and discussions have been incorporated into the revised manuscript (page 7, line 150) and supplementary materials (Figs. S4).

Fig. R15| Cross-sectional SEM image of the perovskite fabricated on top of the control electrode.

Reviewer #3 (Remarks to the Author):

Photodetectors can directly convert optical signals into electrical signals, and play an important role in automatic driving, environmental monitoring, medical imaging, and military fields. Perovskite material has excellent photoelectric performance of direct band gap, large absorption coefficient and electron hole diffusion length, which opens a new door for the research of photodetectors. In this paper, The authors propose a significant electric field modulation strategy, based on which the dark current is significantly reduced and the ion migration is effectively suppressed. However, there are still problems in mechanism innovation and conclusion verification. The details are as follows:

Response: We appreciate for the thoughtful comments and meaningful suggestions. We have carefully read your comments and noticed your concerns. According to your suggestion, we have made corresponding changes in the manuscript. We hope our reply can address your concerns.

***Q1:** In a recent paper by the author, a similar control method was used, that is, the introduction of additional control electrodes to suppress the dark current of the X-ray detector. Compared with this manuscript, what are the innovations in physical mechanism and technical methods?*

Response: Thank you very much for your careful review. In our recent published work, the perovskite/C₆₀/In₂O₃ heterojunction structure was applied. In this manuscript, we developed a simpler device structure with Au electrodes directly contacting with perovskite, and provided a more general physical model to reduce dark current, rendering it more universal and having a wider application prospect. We verified the physical mechanism of the carrier transport process and dark current reduction using electrical field simulation, making this work more convincing and instructive. Furthermore, the ion migration suppression effect of control voltage is first proposed and demonstrated in this work. In terms of the technical methods, we fabricated a flexible active-matrix photodetector array and successfully demonstrated its application in wearable health monitoring and flexible imaging sensors.

Q2: *The author introduces a third electrode on the other side of the photoconductor to suppress the dark current. The introduced third electrode is similar to the gate in the transistor, and What is the difference between both them?*

Response: Thanks for your critical comment. Compared with the field-effect transistor (FET) structure, our control electrode is directly in contact with the semiconductor material, while the gate electrode in a FET is separated from the semiconductor material by a dielectric layer (Fig. R16). Although the current between the source and drain electrodes can be modulated by the gate voltage in the transistor structure, its current value is always positive because the off-state current of the transistor is directly related to the channel resistance. The control electrode in our structure can directly participate in the current transport process between the source and drain electrodes, so it has a stronger control effect on the dark current, and the value of the dark current can be reduced to zero or even a negative value (Fig. 2c). We appreciate for the constructive advice and have added the comments in our revised manuscript (page 3, line 70) and supplementary materials (Figs. S1).

Fig. R16 | Schematic of a bottom gate top contact (BGTC) transistor.

Q3: *The author mentioned that the iodine ion migration was inhibited by introducing the control electrode, but no direct evidence was given to prove this.*

Response: Thanks to the reviewer for pointing out this important point. To demonstrate the effective suppression of ion migration by our method, we characterized the perovskite films by *in situ* photoluminescence (PL) test, since the intensity change of PL can directly prove the occurrence of ion migration [*Nat. Mater.* **14**, 193-198 (2015)] [*Nat. Commun.* **9**, 5113 (2018)]. In addition, the edge of the signal electrode is the region where ion migration is most likely to occur because of its highest electric field strength [*Energy Environ. Sci.* **15**, 5324-5339 (2022)].

Fig. R17a and b show the schematic diagram of measuring the PL intensity of perovskite thin films around signal electrodes using the confocal microscope spectrometer (Alpha300, WITec). After 0.5 V bias for 60 min, the PL intensity of conventional photoconductive-type FPD decreased significantly, while the PL intensity of the electrical field modulated FPD was almost unchanged, which proves that our method has a significant inhibitory effect on ion migration (Fig. R17c,d). The corresponding results and discussions have been incorporated into the revised manuscript (page 6, line 132) and supplementary materials (Figure S3).

Fig. R17 | Measuring the photoluminescence intensity around the signal electrode. a,b, Schematic of the *in situ* PL intensity measurement around signal electrodes of conventional photoconductive-type (a) and electrical field modulated (b) FPDs. **c,d,** Measured PL intensity spectra near the signal electrode edges of photoconductive-type (c) and electrical field modulated (d) FPDs.

Q4: Is the response time related to the control electrode voltage?

Response: Thanks for the valuable comment. The response time of the FPD is indeed related to the control electrode voltage. As shown in Fig. 2b, when the working voltage is constant, the electric field distribution near the signal electrode changes as the control voltage changes. Specifically, as the control electrode voltage increases, the electric field strength near the signal

electrode becomes weaker. Therefore, the introduction of control voltage will inevitably increase the response time because the transit time is proportional to the electric field strength [Appl. Phys. Rev. 7, 011315 (2020)]. When the working voltage is fixed at 0.1 V, the response time of the FPD under same light pulses (1 mW/cm², 520 nm) exhibited increased response time of ~35, 46, and 52 ms, at different control voltages of 0, 0.1, and 0.2 V, as shown in Fig. R18. Relevant discussion and data have been added to the revised manuscript (page 11, line 264) and supplementary material (Fig. S19).

Fig. R18 | Response time of FPD under 0, 0.1, 0.2 V control voltage modulation, when the working voltage is fixed at 0.1 V.

Q5: What is the carrier mobility of the device in the thin film transistor array? Compared with similar devices, does the carrier mobility increase due to the introduction of the third electrode?

Response: Thank you for the inspiring questions. The statistical distributions of mobility (average $\mu_{lin}=0.44 \text{ cm}^2 \text{ V}^{-1} \text{ s}^{-1}$) of 256 In₂O₃ TFTs in the array are shown in Fig. R19a. And the linear mobility (μ_{lin}) of the transistors was extracted with:

$$\mu_{lin} = \frac{L}{WC_{ox}V_{DS}} \times \frac{dI_D}{dV_{GS}}$$

Where W and L are the channel width and length of the transistor, respectively. I_D and V_{GS} represent the drain current and gate voltage, respectively. C_{ox} is the capacitance of the gate dielectric layer.

In order to explore the effect of control voltage (0.1 V) on the mobility of transistors, we measured the transfer characteristic curves of transistors under different conditions. As shown in Fig. R19b, the transfer curve of the transistor is almost independent of the control voltage because the capacitance between the control electrode and the In₂O₃ semiconductor is smaller than that between the gate electrode and the In₂O₃ semiconductor. The capacitance can be calculated by:

$$C = \frac{\epsilon_0 \epsilon_r S}{d}$$

where ϵ_0 is the permittivity of free space, ϵ_r is the relative permittivity, S is the area, and d is the thickness. The thickness of SiO₂ is 100 nm, and the thickness of SU-8 photoresist is 5 μm (Fig. R19c). The dielectric constant of SiO₂ is 3.9 [*Rep. Prog. Phys.* **69**, 327, (2006)] and that of SU-8 is 3.2 (Available: <https://kayakuam.com/products/su-8-2000/>). Therefore, under the same conditions, the ratio of the control electrode capacitance ($C_{Control}$) to the gate capacitance (C_{Gate}) is 1:61. In addition, the voltage of the control electrode was fixed at 0.1 V while the gate voltage was 20 V during the test ($\frac{C_{Control}}{C_{Gate}} = \frac{1}{12200}$). Therefore, the control electrode voltage has almost no effect on the transistor mobility. The relevant discussion has been added in the revised manuscript (page 14, line 332) and supplementary materials (Fig. S28).

Fig. R19 | Effect of control voltage on transistor mobility. **a**, Statistical distributions of the mobility (average $\mu_{lin}=0.44 \text{ cm}^2 \text{ V}^{-1} \text{ s}^{-1}$) of 256 In₂O₃ TFTs. **b**, Transfer curves of the In₂O₃ TFT with and without a control voltage applied. **c**, The thickness of the SU-8 film was $\sim 5 \mu\text{m}$ measured by a stylus profiler (P7, KLA-Tencor).

Q6: In Fig. 3h and Fig. 5f, the reason why the photocurrent rises first and then falls?

Response: Thank you for your careful review and valuable comment. We have noticed the phenomenon that the photocurrent rises first and then falls. This phenomenon could be attributed to the unstable electrical pulse signals generated by the function generator, which is utilized to drive the light source (semiconductor laser).

To confirm this, we did comparison experiments by using a mechanical chopper (SSI-instrument, OE3001) as a modulation method to provide optical pulse inputs and measured the response of the photoconductor, under the same light power density condition (R20a, ii). In this test setup, the photocurrent becomes stable (Fig. R20b).

This proves that the current noise could be attributed to the input electrical pulse signals, rather than the device. Its corresponding explanation and discussion have been added to the revised manuscript (page 11, line 260) and supplementary materials (Fig. S17).

Fig. R20 | The photocurrent of FPD obtained by pulses light illumination with different modulation methods. a, Light pulse generation method and **b,** corresponding photocurrent signal.

Q7: The author shows a response pulse of photoconductivity under extremely weak light intensity in the supporting information Fig. 10. It is suggested that the author increase at least 10 stable response periods to increase credibility.

Response: We thank the referee for this helpful suggestion. The corresponding weak light response data for twenty cycles of the FPD (Fig. R21) has been added to the supplementary material (Fig. S13), and its obvious photocurrent response demonstrates its excellent stability.

Fig. R21 | Time-resolved photocurrent response of FPD under weak light illumination (520 nm, 30 pW/cm²) when the control voltage was 0.1 V. a, Twenty representative switching cycles of the FPD show good stability. **b,** Enlarged response pulse waveform in one period.

Q8: Influence of film quality on voltage selection of control electrode needs to be pointed out.

Response: Thanks for your critical comment. During the experiment, we found that the surface roughness and quality of the perovskite film has a crucial influence on the voltage selection of the control electrode. To study this, we used different spin-coating processes (different dripping times of antisolvent) to prepare two perovskite films, because the antisolvent addition time has a great influence on the quality of perovskite thin films [*Adv. Mater. Interfaces* **7**, 2000950 (2020)] [*Nat. Commun.* **12**,1878 (2021)]. The root mean square (RMS) surface roughness of the perovskite films was measured by atomic force microscopy (AFM, Bruker Dimension ICON). Fig. R22a shows the control film with the RMS surface roughness of 25.7 nm and antisolvent CB added at 5 seconds countdown. However, when the antisolvent CB was added in advance (at 25 s countdown), the film roughness increased significantly (52.9 nm), as shown in Fig. R22b.

To further evaluate the influence of the surface roughness of the perovskite film on the voltage selection of the control electrode, we measured the dark current as a function of the control voltage (working voltage was fixed at 0.1 V). Fig. R22c shows that the dark current is suppressed to near zero when the control voltage is 90 mV (critical voltage < working voltage), because the sharp part of the surface of the rough perovskite film will lead to a higher local electric field than the flat surface, resulting in a stronger control effect [*Appl. Phys. Lett.* **89**,

133506 (2006)] [*Phys. Rev. B* **60**, 9157-9164 (1999)]. Therefore, when the control voltage is set to 0.1 V, the current-time curve of the dark current stabilizes at a negative value (-28 pA), as shown in Fig. R22d.

These results illustrate that the film quality (e.g., surface roughness) could influence the selection of the control voltage, and smaller voltage values could be utilized when film has larger roughness values due to the enhanced local electric field (at sharp region). Relevant comments and results have been added to the revised manuscript (page 8, line 178) and supplementary material (Figure S11).

Fig. R22 | Effect of perovskite film quality on the voltage selection of control electrode. **a,b**, Surface morphologies of perovskite films fabricated with different antisolvent addition times. The root mean square (RMS) surface roughness of perovskite film increases significantly from 25.7 nm (**a**) to 52.9 nm (**b**), when the antisolvent CB addition time changes from 5 s to 25 s countdown. **c,d**, Electrical properties of perovskite films with higher surface roughness. (**c**) Dark current varies with the control voltage when the working voltage is set to 0.1 V (critical voltage=90 mV). (**d**) *I-t* curve of the signal electrode under 0.1 V control voltage (dark current=-28 pA).

REVIEWERS' COMMENTS

Reviewer #1 (Remarks to the Author):

The authors are very attentive to the points raised by the reviewers, adding additional experiments and discussion. The revised paper is very good and deserves publication.

Reviewer #2 (Remarks to the Author):

The authors replied to all my comments, providing new data and explanations. I, therefore, highly recommend the publication of this article without any further delays.

Reviewer #3 (Remarks to the Author):

The submitted manuscript has been carefully revised, providing detailed responses and supplementary experiments to the comments, giving a clearer explanation of the mechanism and a more complete presentation of the results. My suggestion is to accept the revised manuscript.

Responses to reviewers' comments

Reviewer #1 (Remarks to the Author):

The authors are very attentive to the points raised by the reviewers, adding additional experiments and discussion. The revised paper is very good and deserves publication.

Response: We thank the reviewer for the positive comments. We appreciate your great contributions.

Reviewer #2 (Remarks to the Author):

The authors replied to all my comments, providing new data and explanations. I, therefore, highly recommend the publication of this article without any further delays.

Response: Thank you very much for your careful review and suggestions to make the manuscript better. We appreciate your great contributions.

Reviewer #3 (Remarks to the Author):

The submitted manuscript has been carefully revised, providing detailed responses and supplementary experiments to the comments, giving a clearer explanation of the mechanism and a more complete presentation of the results. My suggestion is to accept the revised manuscript.

Response: Thank you very much for your careful review and suggestions to improve our manuscript. We appreciate your great contributions.